# Learnable Embedding Sizes for Recommender Systems

**Siyi Liu[1], Chen Gao[2,]\*, Yihong Chen[3], Depeng Jin[2], Yong Li[2]**
[1]University of Electronic Science and Technology of China, Chengdu, China
[2]Beijing National Research Center for Information Science and Technology,
Department of Electronic Engineering, Tsinghua University, Beijing 100084, China
[3]University College London, London, United Kingdom
`ssui.liu1022@gmail.com, gc16@mails.tsinghua.edu.cn,`
`yihong-chen@outlook.com, {jindp,liyong07}@tsinghua.edu.cn`

## Abstract

The embedding-based representation learning is commonly used in deep learning recommendation models to map the raw sparse features to dense vectors. The traditional embedding manner that assigns a uniform size to all features has two issues. First, the numerous features inevitably lead to a gigantic embedding table that causes a high memory usage cost. Second, it is likely to cause the over-fitting problem for those features that do not require too large representation capacity. Existing works that try to address the problem always cause a significant drop in recommendation performance or suffer from the limitation of unaffordable training time cost. In this paper, we propose a novel approach, named PEP[1] (short for **P**lug-in **E**mbedding **P**runing), to reduce the size of the embedding table while avoiding the drop of recommendation accuracy. PEP prunes embedding parameter where the pruning threshold(s) can be adaptively learned from data. Therefore we can automatically obtain a mixed-dimension embedding-scheme by pruning redundant parameters for each feature. PEP is a general framework that can plug in various base recommendation models. Extensive experiments demonstrate it can efficiently cut down embedding parameters and boost the base model's performance. Specifically, it achieves strong recommendation performance while reducing 97-99% parameters. As for the computation cost, PEP only brings an additional 20-30% time cost compared with base models.

## 1 Introduction

The success of deep learning-based recommendation models (Zhang et al., 2019) demonstrates their advantage in learning feature representations, especially for the most widely-used categorical features. These models utilize the embedding technique to map these sparse categorical features into real-valued dense vectors to extract users' preferences and items' characteristics. The learned vectors are then fed into prediction models, such as the inner product in FM (Rendle, 2010), self-attention networks in AutoInt (Song et al., 2019), to obtain the prediction results. The embedding table could contain a large number of parameters and cost huge amounts of memory since there are always a large number of raw features. Therefore, the embedding table takes the most storage cost.

A good case in point is the YouTube Recommendation Systems (Covington et al., 2016). It demands tens of millions of parameters for embeddings of the YouTube video IDs. Considering the increasing demand for instant recommendations in today's service providers, the scale of embedding tables becomes the efficiency bottleneck of deep learning recommendation models. On the other hand, features with uniform embedding size may hard to handle the heterogeneity among different features. For example, some features are more sparse, and assigning too large embedding sizes is likely

---

\*Chen Gao is the Corresponding Author. The work is performed when Siyi Liu is an intern in Tsinghua University.

[1]Codes are available at: `https://github.com/ssui-liu/learnable-embed-sizes-for-RecSys`

to result in over-fitting issues. Consequently, recommendation models tend to be sub-optimal when embedding sizes are uniform for all features.

The existing works towards this problem can be divided into two categories. Some works (Zhang et al., 2020; Shi et al., 2020; Kang et al., 2020) proposed that some closely-related features can share parts of embeddings, reducing the whole cost. Some other works (Joglekar et al., 2020; Zhao et al., 2020b;a; Cheng et al., 2020) proposed to assign embeddings with flexible sizes to different features relying on human-designed rules (Ginart et al., 2019) or neural architecture search (Joglekar et al., 2020; Zhao et al., 2020b;a; Cheng et al., 2020). Despite a reduced embedding size table, these methods still cannot perform well on the two most concerned aspects, recommendation performance and computation cost. Specifically, these methods either obtain poor recommendation performance or spend a lot of time and efforts in getting proper embedding sizes.

In this paper, to address the limitations of existing works, we proposed a simple yet effective pruning-based framework, named **P**lug-in **E**mbedding **P**runing (PEP), which can plug in various embedding-based recommendation models. Our method adopts a direct manner–pruning those unnecessary embedding parameters in one shot–to reduce parameter number.

Specifically, we introduce the learnable threshold(s) that can be jointly trained with embedding parameters via gradient descent. Note that the threshold is utilized to determine the importance of each parameter *automatically*. Then the elements in the embedding vector that are smaller than the threshold will be pruned. Then the whole embedding table is pruned to make sure each feature has a suitable embedding size. That is, the embedding sizes are flexible. After getting the pruned embedding table, we retrain the recommendation model with the inspiration of the Lottery Ticket Hypothesis (LTH) (Frankle & Carbin, 2018), which demonstrates that a subnetwork can reach higher accuracy compared with the original network. Based on flexible embedding sizes and the LTH, our PEP can cuts down embedding parameters while maintaining and even boosting the model's recommendation performance. Finally, while there is always a trade-off between recommendation performance and parameter number, our PEP can obtain multiple pruned embedding tables by running only once. In other words, our PEP can generate several memory-efficient embedding matrices once-for-all, which can well handle the various demands for performance or memory-efficiency in real-world applications. We conduct extensive experiments on three public benchmark datasets: Criteo, Avazu, and MovieLens-1M. The results demonstrate that our PEP can not only achieve the best performance compared with state-of-the-art baselines but also reduces 97% to 99% parameter usage. Further studies show that our PEP is quite computationally-efficient, requiring a few additional time for embedding-size learning. Furthermore, visualization and interpretability analysis on learned embedding confirm that our PEP can capture features' intrinsic properties, which provides insights for future researches.

## 2 RELATED WORK

Existing works try to reduce the embedding table size of recommendation models from two perspectives, embedding parameter sharing and embedding size selection.

### 2.1 EMBEDDING PARAMETER SHARING

The core idea of these methods is to make different features re-use embeddings via parameter sharing. Kang et al. (2020) proposed MGQE that retrieves embedding fragments from a small size of shared centroid embeddings and then generates final embedding by concatenating those fragments. Zhang et al. (2020) used the double-hash trick to make low-frequency features share a small embedding-table while reducing the likelihood of a hash collision. Shi et al. (2020) tried to yield a unique embedding vector for each feature category from a small embedding table by combining multiple smaller embedding (called embedding fragments). The combination is usually through concatenation, add, or element-wise multiplication among embedding fragments.

However, those methods suffer from two limitations. First, engineers are required to carefully design the parameter-sharing ratio to balance accuracy and memory costs. Second, these rough embedding-sharing strategies cannot find the redundant parts in the embedding tables, and thus it always causes a drop in recommendation performance.

Table 1: Comparison of our PEP and existing works (AutoInt is a base recommendation model and others are embedding-parameter-reduction methods.)

| Method | Performance | Parameter Number | Computation Cost |
|---|---|---|---|
| AutoInt (Song et al., 2019) | √ | × | √ |
| MDE (Ginart et al., 2019) | × | √ | × |
| NIS (Joglekar et al., 2020) | × | √ | × |
| DartsEmb (Zhao et al., 2020b) | √ | √ | × |
| DNIS (Cheng et al., 2020) | √ | √ | × |
| Our PEP | √ | √ | √ |

In this work, our method automatically chooses suitable embedding usages by learning from data. Therefore, engineers can be free from massive efforts for designing sharing strategy, and the model performance can be boosted via removing redundant parameters and alleviating the over-fitting issue.

## 2.2 EMEBDDING SIZE SELECTION

The embedding-sharing methods assign uniform embedding sizes to every feature, which may still fail to deal with the heterogeneity among different features. Recently, several methods proposed a new paradigm of mixed-dimension embedding table. Specifically, different from assigning all features with uniformed embedding size, different features can have different embedding sizes. MDE (Ginart et al., 2019) proposed a human-defined rule that the embedding size of a feature is proportional to its popularity. However, this rule-based method is too rough and cannot handle those important features with low frequency. Additionally, there are plenty of hyper-parameters in MDE requiring a lot of truning efforts. Some other works (Joglekar et al., 2020; Zhao et al., 2020b;a; Cheng et al., 2020) assigned adaptive embedding sizes to different features, relying on the advances in Neural Architecture Search (NAS) (Elsken et al., 2019), a significant research direction of Automated Machine Learning (AutoML) (Hutter et al., 2019). NIS (Joglekar et al., 2020) used a reinforcement learning-based algorithm to search embedding size from a candidate set predefined by human experts. A controller is adopted to generate the probability distribution of size for specific feature embeddings. This was further extended by DartsEmb (Zhao et al., 2020b) by replacing the reinforcement learning searching algorithm with differentiable search (Liu et al., 2018). AutoDim (Zhao et al., 2020a) allocated different embedding sizes for different feature fields, rather than individual features, in a same way as DartsEmb. DNIS (Cheng et al., 2020) made the candidate embedding size to be continuous without predefined candidate dimensions. However, all these NAS-based methods require extremely high computation costs in the searching procedure. Even for methods that adopt differential architecture search algorithms, the searching cost is still not affordable. Moreover, these methods also require a great effort in designing proper search spaces.

Different from these works, our pruning-based method can be trained quite efficiently and does not require any human efforts in determining the embedding-size candidates.

## 3 PROBLEM FORMULATION

Feature-based recommender system[2] is commonly used in today's information services. In general, deep learning recommendation models take various raw features, including users' profiles and items' attributes, as input and predict the probability that a user like an item. Specifically, models take the combination of user's profiles and item's attributes, denoted by $\mathbf{x}$, as its' input vector, where $\mathbf{x}$ is the concatenation of all fields that could defined as follows:

$$\mathbf{x} = [\mathbf{x_1}; \mathbf{x_2}; \dots; \mathbf{x_M}], \tag{1}$$

where $\mathbf{M}$ denotes the number of total feature fields, and $\mathbf{x_i}$ is the feature representation (one-hot vector in usual) of the $i$-th field. Then for $\mathbf{x_i}$, the embedding-based recommendation models generate corresponding embedding vector $\mathbf{v_i}$ via following formulation:

$$\mathbf{v_i} = \mathbf{V_i}\mathbf{x_i}, \tag{2}$$

---

[2]It is also known as click-through rate prediction.

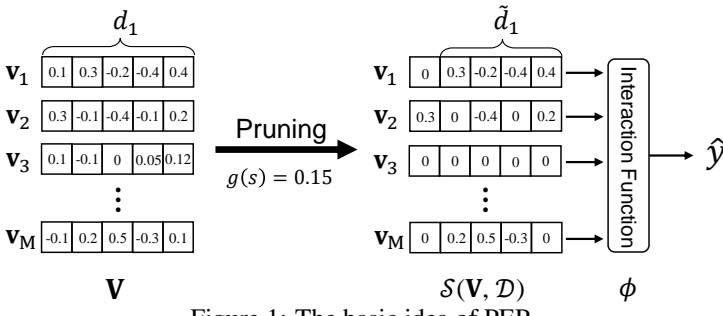

Figure 1: The basic idea of PEP.

where $\mathbf{V_i} \in \mathrm{R}^{n_i \times d}$ is an embedding matrix of $i$-th field, $n_i$ denotes the number of features in the $i$-th field, and $d$ denotes the size of embedding vectors. The model's embedding matrices $\mathbf{V}$ for all fields of features can be formulated as follows,

$$\mathbf{V} = \{\mathbf{V_1}, \mathbf{V_2}, \dots, \mathbf{V_M}\}, \tag{3}$$

The prediction score could be calculated with $\mathbf{V}$ and model's other parameters (mainly refer to the parameters in prediction model) $\Theta$ as follows,

$$\hat{y} = \phi(\mathbf{x}|\mathbf{V}, \Theta), \tag{4}$$

where $\hat{y}$ is the predicted probability and $\phi$ represent the prediction model, such as FM (Rendle, 2010) or AutoInt (Song et al., 2019). As for model training, to learn the models parameters, the optimizer minimizes the training loss as follows,

$$\min \mathcal{L}(\mathbf{V}, \Theta, \mathcal{D}), \tag{5}$$

where $\mathcal{D} = \{\mathbf{x}, y\}$ represents the data fed into the model, $\mathbf{x}$ denotes the input feature, $y$ denotes the ground truth label, and $\mathcal{L}$ is the loss function. The Logloss is the most widely-used loss function in recommendation tasks (Rendle, 2010; Guo et al., 2017; Song et al., 2019) and calculated as follows,

$$\mathcal{L} = -\frac{1}{|\mathcal{D}|} \sum_{j=1}^{|\mathcal{D}|} \left( y_j \log\left(\hat{y}_j\right) + (1 - y_j) \log\left(1 - \hat{y}_j\right) \right), \tag{6}$$

where $|\mathcal{D}|$ is the total number of training samples and regularization terms are omitted for simplification.

## 4 METHODOLOGY

### 4.1 LEARNABLE EMBEDDING SIZES THROUGH PRUNING

As mentioned above, a feasible solution for memory-efficient embedding learning is to automatically assign different embedding sizes $\tilde{d}_i$ for different features embeddings $\mathbf{v}_i$, which is our goal. However, to learn $\tilde{d}_i$ directly is infeasible due to its discreteness and extremely-large optimization space. To address it, we propose a novel idea that enforce column-wise sparsity on $\mathbf{V}$, which equivalently shrinks the embedding size. For example, as it shown in Figure 1, the first value in embedding $\mathbf{v}_1$ is pruned and set to zero, leading to a $\tilde{d}_1 = d_1 - 1$ embedding size in effect. Furthermore, some unimportant feature embeddings, like $\mathbf{v}_3$, are dropped by set all values to zero[3]. Thus our method can significantly cut down embedding parameters. Note that the technique of *sparse matrix storage* help us to significantly save memory usage (Virtanen et al., 2020).

In such a way, we recast the problem of embedding-size selection into learning column-wise sparsity for the embedding matrix $\mathbf{V}$. To achieve that, we design a sparsity constraint on $\mathbf{V}$ as follows,

$$\min \mathcal{L}, s.t. ||\mathbf{V}||_0 \leq k, \tag{7}$$

where $||\cdot||_0$ denotes the $L_0$-norm, *i.e.* the number of non-zeros and $k$ is the parameter budget, which is, the constraint on the total number of embedding parameters.

---

[3]Our PEP benefit from such kind of reduction, as demonstrated in Section 5.1, 5.3 and 5.4.

However, direct optimization of Equation (7) is NP-hard due to the non-convexity of the $L_0$-norm constraint. To solve this problem, the convex relaxation of $L_0$-norm, called $L_1$-norm, has been studied for a long time (Taheri & Vorobyov, 2011; Beck & Teboulle, 2009; Jain et al., 2014). For example, the Projected Gradient Descent (PGD) (Jain et al., 2014) in particular has been proposed to project parameters to $L_1$ ball to make the gradient computable in almost closed form. Note that the $L_1$ ball projection is also known as Soft Thresholding (Kusupati et al., 2020). Nevertheless, such methods are still faced with two major issues. First, the process of projecting the optimization values onto $L_1$ ball requires too much computation cost, especially when the recommendation model has millions of parameters. Second, the parameter budget $k$ requires human experts to manually set at a global level. Considering that features have various importance for recommendation, such operation is obviously sub-optimal. To tackle those two challenges, inspired by Soft Threshold Reparameterization (Kusupati et al., 2020), we directly optimize the projection of $\mathbf{V}$ and adaptively pruning the $\mathbf{V}$ via learnable threshold(s) which can be updated by gradient descent. The re-parameterization of $\mathbf{V}$ can be formulated as follows,

$$\hat{\mathbf{V}} = \mathcal{S}(\mathbf{V}, s) = sign(\mathbf{V})\text{ReLU}(|\mathbf{V}| - g(s)), \tag{8}$$

where $\hat{\mathbf{V}} \in \mathcal{R}^{N \times d}$ denotes the re-parameterized embedding matrix, and $g(s)$ serves as a pruning threshold value, of which *sigmoid* function is a simple yet effective solution.[4] We set the initial value of trainable parameter $s \in \mathcal{R}$ (called $s_{\text{init}}$) to make sure that the threshold(s) $g$ start close to zero. The $sign(\cdot)$ function converts positive input value to 1 and negative input value to -1, and zero input will keep unchanged.

As $\mathcal{S}(\mathbf{V}, s)$ is applied to each element of $\mathbf{V}$, and thus the optimization problem in Equation (5) could be redefined as follows,

$$\min \mathcal{L}(\mathcal{S}(\mathbf{V}, s), \Theta, \mathcal{D}). \tag{9}$$

Then the trainable pruning parameter $s$ could be jointly optimized with parameters of the recommendation models $\phi$, through the standard back-propagation. Specifically, the gradient descent update equation for $\mathbf{V}$ at $t$-th step is formulated as follows,

$$\mathbf{V}^{(t+1)} \leftarrow \mathbf{V}^{(t)} - \eta_t \nabla_{\mathcal{S}(\mathbf{V}, s)} \mathcal{L}\left(\mathcal{S}(\mathbf{V}^{(t)}, s), \mathcal{D}\right) \odot \nabla_{\mathbf{V}} \mathcal{S}(\mathbf{V}, s), \tag{10}$$

where $\eta_t$ is $t$-th step learning rate and $\odot$ denotes the Hadamard product. To solve the non-differentiablilty of $\mathcal{S}(\cdot)$, we use sub-gradient to reformat the update equation as follows,

$$\mathbf{V}^{(t+1)} \leftarrow \mathbf{V}^{(t)} - \eta_t \nabla_{\mathcal{S}(\mathbf{V}, s)} \mathcal{L}\left(\mathcal{S}(\mathbf{V}^{(t)}, s), \mathcal{D}\right) \odot \mathbf{1}\left\{\mathcal{S}(\mathbf{V}^{(t)}, s) \neq 0\right\}, \tag{11}$$

where $\mathbf{1}\{\cdot\}$ denotes the indicator function. Then, as long as we choose a continuous function $g$ in $\mathcal{S}(\cdot)$, then the loss function $\mathcal{L}\left(\mathcal{S}(\mathbf{V}^{(t)}, s), \mathcal{D}\right)$ would be continuous for $s$. Moreover, the sub-gradient of $\mathcal{L}$ with respect to $s$ can be used of gradient descent on $s$ as well.

Thanks to the automatic differentiation framework like TensorFlow (Abadi et al., 2016) and PyTorch (Paszke et al., 2019), we are free from above complex gradient computation process. Our PEP code can be found in Figure 7 of Appendix A.2. As we can see, it is quite simple to incorporate with existing recommendation models, and there is no need for us to manually design the back-propagation process.

## 4.2 RETRAIN WITH LOTTERY TICKET HYPOTHESIS

After pruning the embedding matrix $\mathbf{V}$ to the target parameter budget $\mathcal{P}$, we could create a binary pruning mask $m \in \{0, 1\}^{\hat{\mathbf{V}}}$ that determines which parameter should remain or drop. Then we retrain the base model with a pruned embedding table. The Lottery Ticket Hypothesis (Frankle & Carbin, 2018) illustrates that a sub-network in a randomly-initialized dense network can match the original network, when trained in isolation in the same number of iterations. This sub-network is called the *winning ticket*. Hence, instead of randomly re-initializing the weight, we retrain the base model while re-initializing the weights back to their original (but masked now) weights $m \odot \mathbf{V}_0$. This initiation strategy can make the training process faster and stable, keeping the performance consistent, which is shown in Appendix A.6.

---

[4]More details about how to choose a suitable $g(s)$ are provided in Appendix A.1.

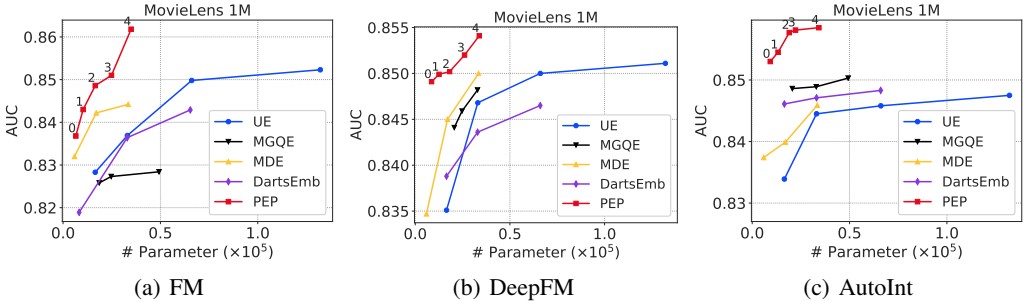

Figure 2: AUC-# Parameter curve on MovieLens-1M with three base models.

### 4.3 PRUNING WITH FINER GRANULARITY

Threshold parameter $s$ in Equation (8) is set to a scalar that values of every dimension will have the same threshold value. We name this version as *global wise pruning*. However, different dimensions in the embedding vector $\mathbf{v}_i$ may have various importance, and different fields of features may also have highly various importance. Thus, values in the embedding matrix require different sparsity budgets, and pruning with a global threshold may not be optimal. To better handle the heterogeneity among different features/dimensions in $\mathbf{V}$, we design following different threshold tactic with different granularities. (1) *Dimension Wise*: The threshold parameter $s$ is set as a vector $\mathbf{s} \in \mathcal{R}^d$. Each value in an embedding will be pruned individually. (2) *Feature Wise:* The threshold parameter $s$ is defined as a vector $\mathbf{s} \in \mathcal{R}^N$. Pruning on each features' embedding could be done in separate ways. (3) *Feature-Dimension Wise:* this variant combines the above genre of threshold to obtain the finest granularity pruning. Specifically, thresholds are set as a matrix $\mathbf{s} \in \mathcal{R}^{N \times d}$.

## 5 EXPERIMENTS

**Dataset**. We use three benchmark datasets: MovieLens-1M, Criteo, and Avazu, in our experiments.

**Metric**. We adopt AUC (Area Under the ROC Curve) and Logloss to measure the performance of models.

**Baselines and Base Recommendation Models**. We compared our PEP with traditional UE (short for Uniform Embedding). We also compare with the recent advances in flexible embedding sizes: MGQE (Kang et al., 2020), MDE (Ginart et al., 2019), and DartsEmb (Zhao et al., 2020b)[5]. We deploy PEP and all baseline methods to three representative feature-based recommendation models: **FM** (Rendle, 2010), **DeepFM** (Guo et al., 2017), and **AutoInt** (Song et al., 2019), to compare their performance[6].

### 5.1 RECOMMENDATION ACCURACY AND PARAMETER NUMBER

We present the curve of recommendation performance and parameter number in Figure 2, 3 and 4, including our method and state-of-the-art baseline methods. Since there is a trade-off between recommendation performance and parameter number, the curves are made of points that have different sparsity demands[7].

- **Our method reduces the number of parameters significantly**. Our PEP achieves the highest reduce-ratio of parameter number in all experiments, especially in relatively large datasets (Criteo and Avazu). Specifically, in Criteo and Avazu datasets, our PEP-0 can reduce 99.90% parameter usage compared with the best baseline (from the $10^6$ level to the $10^3$ level, which is very significant.). Embedding matrix with such low parameter usage means that only hundreds of embeddings are non-zero. By setting less-important features' embedding to zero, our PEP can break the limitation in existing methods that minimum embedding size is one rather than zero. We conduct more analysis on the MovieLens dataset in Section 5.3 and 5.4 to help us understand why our method can achieve such an effective parameter decreasing.

---

[5]We do not compare with NIS (Joglekar et al., 2020) since it has not released codes and its reinforcement-learning based search is really slow.

[6]More details of implementation and above information could be found in Appendix A.4.

[7]We report five points of our method, marked from 0 to 4.

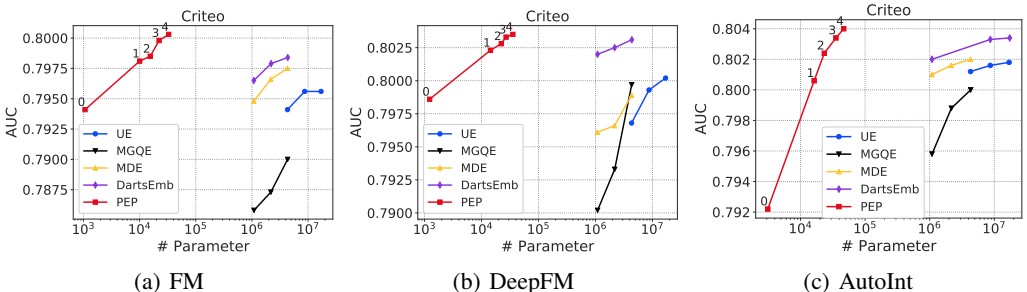

Figure 3: AUC-# Parameter curve on Criteo with three base models.

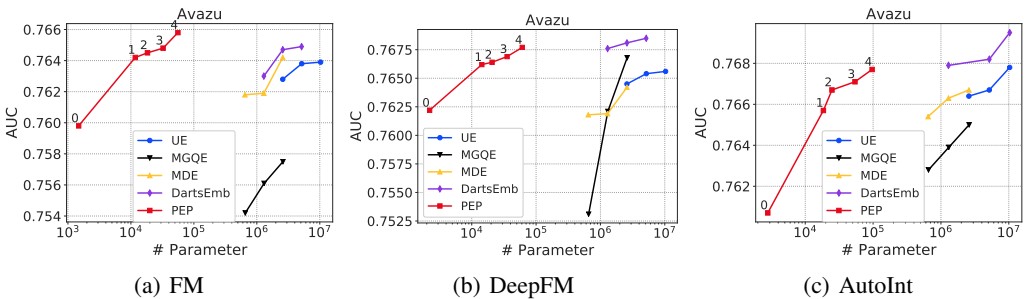

Figure 4: AUC-# Parameter curve on Avazu with three base models.

- **Our method achieves strong recommendation performance.** Our method consistently outperforms the uniform embedding based model and achieves better accuracy than other methods in most cases. Specifically, for the FM model on the Criteo dataset, the relative performance improvement of PEP over UE is 0.59% and over DartsEmb is 0.24% in terms of AUC. Please note that the improvement of AUC or Logloss at such level is still considerable for feature-based recommendation tasks (Cheng et al., 2016; Guo et al., 2017), especially considering that we have reduced a lot of parameters. A similar improvement can also be observed from the experiments on other datasets and other recommendation models. It is worth noting that our method could keep a strong AUC performance under extreme sparsity-regime. For example, when the number of parameters is only in the $10^3$ level (a really small one), the recommendation performance still remarkably outperforms the Linear Regression model (more details can be found in Appendix A.5).

To summarize it, with the effectiveness of recommendation accuracy and parameter-size reduction, the PEP forms a frontier curve encompassing all the baselines at all the levels of parameters. This verifies the superiority that our method can handle different parameter-size budgets well.

## 5.2 EFFICIENCY ANALYSIS OF OUR METHOD

As is shown in Section 5.1, learning a suitable parameter budget can yield a higher-accuracy model while reducing the model's parameter number. Nevertheless, it will induce additional time to find apposite sizes for different features. In this section, we study the computational cost and compare the runtime of each training epoch between PEP and DartsEmb on the Criteo dataset. We implement both models with the same batch size and test them on the same platform.

The training time of each epoch on three different models is given in Table 2. We can observe that our PEP's additional computation-cost is only 20% to 30%, which is acceptable compared with the base model. DartsEmb, however, requires nearly double computation time to search a good embedding size in its bi-level optimization process. Furthermore, DartsEmb needs to search multiple times to fit different memory budgets, since each one requires a complete re-running. Different from DartsEmb, our PEP can obtain several embedding schemes, which can be applied in different application scenarios, in only a single running. As a result, our PEP's time cost on embedding size search can be further reduced in real-world systems.

Table 2: Runtime of each training epoch on Criteo between base model, DartsEmb, and our PEP.

| Runtime (Second) | FM | DeepFM | AutoInt | Avg. time increase |
|---|---|---|---|---|
| **Base Model** | 1,039 | 1,222 | 1,642 | 0 |
| **DartsEmb** | 2,239 | 2,285 | 3,154 | 98.02% |
| **PEP** | 1,341 | 1,525 | 1,963 | 24.47% |

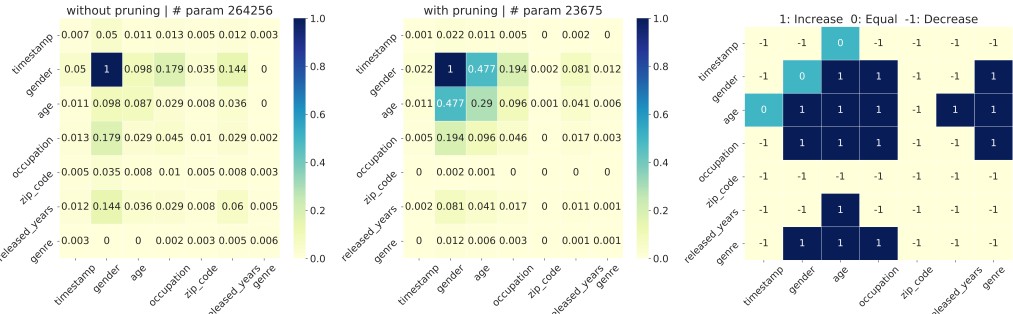

(a) $\mathbf{VV}^\top$ on original embedding    (b) $\mathbf{VV}^\top$ on sparse embedding    (c) Variation of matrix values between original and sparse embedding

Figure 5: Interpretable analysis on MovieLens-1M dataset.

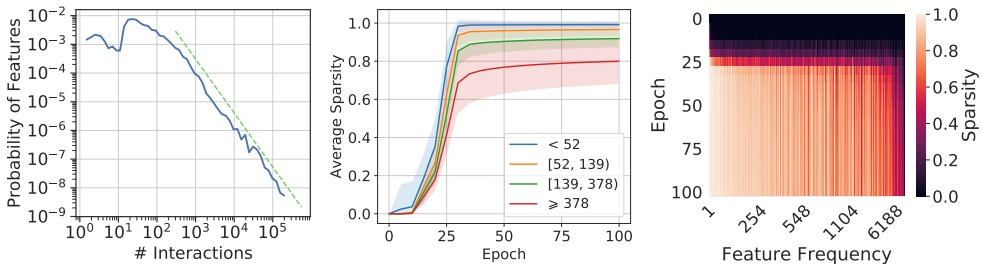

(a) PDF of feature frequencies of MovieLens-1M dataset    (b) Sparsity trajectory generated by PEP on FM    (c) Sparsity heatmap generated by PEP on FM

Figure 6: Correlation between Sparsity and Frequency.

## 5.3 INTERPRETABLE ANALYSIS ON PRUNED EMBEDDINGS

The feature-based recommendation models usually apply the embedding technique to capture two or high order feature interactions. But how does our method work on features interactions? Does our method improve model performance by reducing noisy feature interactions? In this section, we conduct an interpretable analysis by visualizing the feature interaction matrix, calculated by $\mathbf{VV}^\top$. Each value in the matrix is the normalized average of the absolute value of those two field features' dot product result, of which the higher indicates those two fields have a stronger correlation.

Figure 5 (a) and 5 (b) illustrate the interaction matrix without and with pruning respectively, and 5 (c) shows the variation of matrix values. We can see that our PEP can reduce the parameter number between unimportant field interaction while keeping the significance of those meaningful field features' interactions. By denoising those less important feature interactions, the PEP can reduce embedding parameters while maintaining or improving accuracy.

## 5.4 CORRELATION BETWEEN SPARSITY AND FREQUENCY

As is shown in Figure 6 (a), feature frequencies among different features are highly diversified. Thus, using embeddings with uniform size may not handle their heterogeneity, and this property play an important role in embedding size selection. Hence, some recent works (Zhao et al., 2020b; Ginart et al., 2019; Cheng et al., 2020; Kang et al., 2020; Zhang et al., 2020; Joglekar et al., 2020) explicitly utilize the feature frequencies. Different from them, our PEP shrinks the parameter in an end-to-end automatic way, thus circumvents the complex human manipulation. Nevertheless, the

frequency of features is one of the factors that determines whether one feature is important or not. Thus, we study whether our method can detect the influence of frequencies and whether the learned embedding sizes are relevant to the frequency.

We first analyze the sparsity[8] trajectory during training, which is shown in Figure 6 (b), where different colors indicate different groups of features divided according to their popularity. For each group, we first calculate each feature's sparsity, then compute the average on all features. Shades in pictures represent the variance within a group. We can observe that PEP tends to assign high-frequency features larger sizes to make sure there is enough representation capacity. For low-frequency features, the trends are on the contrary. These results are accord to the postulation that high-frequency features deserve more embedding parameters while a few parameters are enough for low-frequency feature embeddings.

Then we probe the relationship between the sparsity of pruned embedding and frequencies of each feature. From Figure 6 (c), we can observe that the general relationship is concord with the above analysis. However, as we can see, some low-frequency features are assigned rich parameters, and some features with larger popularity are assigned small embedding size. This illustrates that simply allocating more parameters to high-frequency features, as most previous works do, can not handle the complex connection between features and their popularities. Our method performs pruning based on data, which can reflect the feature intrinsic proprieties, and thus can cut down parameters in a more elegant and efficient way.

## 6 CONCLUSION

In this paper, we approach the common problem of fixed-size embedding table in today's feature-based recommender systems. We propose a general plug-in framework to learn the suitable embedding sizes for different features adaptively. The proposed PEP method is efficient can be easily applied to various recommendation models. Experiments on three state-of-the-art recommendation models and three benchmark datasets verify that PEP can achieve strong recommendation performance while significantly reducing the parameter number and can be trained efficiently.

## 7 ACKNOWLEDGEMENTS

This work was supported in part by The National Key Research and Development Program of China under grant 2020AAA0106000, the National Natural Science Foundation of China under U1936217, 61971267, 61972223, 61941117, 61861136003.

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

## A  APPENDIX

### A.1  DESCRIPTION OF $g(s)$

Following Kusupati et al. (2020), a proper threshold function $g(s)$ should have following three properties:

1.
$$g(s) > 0, \lim_{s \to -\infty} g(s) = 0, \text{and} \lim_{s \to \infty} g(s) = \infty.$$

2.
$$\exists G \in \mathbb{R}_{++} \ni 0 < g'(s) \le G \ \forall s \in \mathbb{R}.$$

3.
$$g'(s_{\text{init}}) < 1 \text{ which reduce the updating speed of } s \text{ at the initial pruning.}$$

### A.2  PYTORCH CODE OF PEP

We present the main codes of PEP here since it is really easy-to-use and can plug in various embedding-based recommendation models.

```python
import torch
import torch.nn as nn
import torch.nn.functional as F

class PEP(nn.Module):
    # use PEP to replace the original embedding module (torch.nn.Embedding)
    def __init__(self, opt):
        super(PEP, self).__init__()
        self.threshold_type = opt['threshold_type']  # define the granularity of s
        self.latent_dim = opt['latent_dim']          # define the initial embedding size d
        self.feature_num = sum(opt['field_dims'])    # total numbers of features
        init = opt['threshold_init']                 # initial value of s
        self.g = torch.sigmoid                       # define the threshold function g
        self.s = self.init_threshold(init)

        # initialize the embedding matrix V
        self.v = nn.Parameter(torch.rand(self.feature_num, self.latent_dim))
        nn.init.xavier_uniform_(self.v)
        self.sparse_v = self.v.data

    def init_threshold(self, init):  # initialize threshold with different granularities
        if self.threshold_type == 'dimension':
            s = nn.Parameter(init * torch.ones([self.latent_dim]))
        elif self.threshold_type == 'feature':
            s = nn.Parameter(init * torch.ones([self.feature_num, 1]))
        elif self.threshold_type == 'feature_dimension':
            s = nn.Parameter(init * torch.ones([self.feature_num, self.latent_dim]))
        else:
            s = nn.Parameter(init * torch.ones(1))  # set as global by default
        return s

    def soft_threshold(self, v, s):  # the reparameterization function S(V, D)
        return torch.sign(v) * torch.relu(torch.abs(v)-self.g(s))

    def forward(self, x):
        self.sparse_v = self.soft_threshold(self.v, self.s)  # the pruned sparse embedding matrix
        xv = F.embedding(x, self.sparse_v)  # retrieving feature' embeddings by given feature x
        return xv
```

Figure 7: PyTorch code of PEP.

### A.3  WHOLE PROCESS OF PEP

We summarizes the pruning and retrain process by Algorithm 1.

---

**Algorithm 1** Our PEP

---

**Input:** Initial embedding $\mathbf{V}^{(0)}$, base model $\phi$, and target parameter $\mathcal{P}$.
**Output:** Well trained sparsity embedding $\mathbf{V}$.
 1: **while** do not reach $\mathcal{P}$ **do**
 2:     Pruning $\mathbf{V}$ through Equation 9.
 3: **end while**
 4: Obtain binary pruning mask $m = \mathbf{1}\{\mathbf{V}^{(t)}\}$.
 5: Reset the remaining embedding parameter to initial values.
 6: **while** do not coverage **do**
 7:     Minimize the training loss $\mathcal{L}(\mathbf{V}^{(0)} \odot m, \mathcal{D})$ with SGD.
 8: **end while**

---

Table 3: Statistics of three utilized benchmark datasets.

| Dataset | # Samples | # Fields | # Features |
|---|---|---|---|
| MovieLens-1M | $739,015$ | $7$ | $3,864$ |
| Criteo | $45,840,617$ | $39$ | $1,086,810$ |
| Avazu | $40,400,000$ | $22$ | $645,394$ |

## A.4 EXPERIMENTAL SETUP

### A.4.1 DATASETS

We experiment with three public benchmark datasets: MovieLens-1M, Criteo, and Avazu. Table 3 summarizes the statistics of datasets.

- **MovieLens-1M**[9]. It is a widely-used benchmark dataset and contains timestamped user-movie ratings ranging from $1$ to $5$. Following AutoInt (Song et al., 2019), we treat samples with a rating $1, 2$ as negative samples and samples with a rating $4, 5$ as positive samples. Other samples will be treat as neutral samples and removed.

- **Criteo**[10]. This is a benchmark dataset for feature-based recommendation task, which contains 26 categorical feature fields and 13 numerical feature fields. It has about 45 million users' clicking records on displayed ads.

- **Avazu**[11]. Avazu dataset contains 11 days' user clicking behaviors which are released for the Kaggle challenge, There are 22 categorical feature fields in the dataset, and parts of the fields are anonymous.

**Preprocessing** Following the general preprocessing steps (Guo et al., 2017; Song et al., 2019), for numerical feature fields in Criteo, we employ the log transformation of $log^2(x)$ *if* $x > 2$ proposed by the winner of Criteo Competition[12] to normalize the numerical features. Besides, we consider features of which the frequency is less than ten as unknown and treat them as a single feature "*unknown*" for Criteo and Avazu datasets. For each dataset, all the samples are randomly divided into training, validation, and testing set based on the proportion of $80\%$, $10\%$, and $10\%$.

### A.4.2 PERFORMANCE MEASURES

We evaluate the performance of PEP with the following two metrics:

- **AUC**. The area under the Receiver Operating Characteristic or ROC curve (AUC) means the probability to rank a randomly chosen positive sample higher than a randomly chosen negative sample. A model with higher AUC indicates the better performance of the model.

---

[9]https://grouplens.org/datasets/movielens
[10]https://www.kaggle.com/c/criteo-display-ad-challenge
[11]https://www.kaggle.com/c/avazu-ctr-prediction
[12]https://www.csie.ntu.edu.tw/r01922136/kaggle-2014-criteo.pdf

- **Logloss**. As a loss function widely used in the feature-based recommendation, Logloss on test data can straight way evaluate the model's performance. The lower the model's Logloss, the better the model's performance.

### A.4.3 BASELINES

We compared our proposed method with the following state-of-the-art methods:

- **UE** (short for Uniform Embedding). The uniform-embedding manner is commonly accepted in existing recommender systems, of which all features have uniform embedding sizes.

- **MGQE** (Kang et al., 2020). This method retrieves embedding fragments from a small size of shared centroid embeddings, and then generates final embedding by concatenating those fragments. MGQE learns embeddings with different capacities for different items. This method is the most strongest baseline among embedding-parameter-sharing methods.

- **MDE** (short for Mixed Dimension Embedding (Ginart et al., 2019)). This method is based on human-crafted rule, and the embedding size of a specific feature is proportional to its popularity. Higher-frequency features will be assigned larger embedding sizes. This is the state-of-the-art human-rule-based method.

- **DartsEmb** (Zhao et al., 2020b). This is the state-of-the-art neural architecture search-based based method which allows features to automatically search for the embedding sizes in a given space.

### A.4.4 IMPLEMENTATION DETAILS

Following AutoInt (Song et al., 2019) and DeepFM (Guo et al., 2017), we employ Adam optimizer with the learning rate of 0.001 to optimize model parameters in both the pruning and re-training stage. For $g(s)$, we apply $g(s) = \frac{1}{1+e^{-s}}$ in all experiments and initialize the $s$ to $-15$, $-150$ and $-150$ in MovieLens-1M, Criteo and Avazu datasets respectively. Moreover, the granularity of PEP is set as Dimension-wise for PEP-2, PEP-3, and PEP-4 on Criteo and Avazu datasets. And others are set as Feature Dimension-wise. The base embedding dimension $d$ is set to 64 for all the models before pruning. We deploy our method and other baseline methods to three state-of-the-art models: **FM** (Rendle, 2010), **DeepFM** (Guo et al., 2017), and **AutoInt** (Song et al., 2019), to compare their performance. Besides, in the retrain stage, we exploit the early-stopping technique according to the loss of validation dataset during training. We use PyTorch (Paszke et al., 2019) to implement our method and train it with mini-batch size 1024 on a single 12G-Memory NVIDIA TITAN V GPU.

**Implementation of Baseline** For Uniform Embedding, we test the embedding size varying from $[8, 16, 32, 64]$, for the MovieLens-1M dataset. For Criteo and Avazu dataset, we vary the embedding size from $[4, 8, 16]$ because performance starts to drop when $d > 16$.

For other baseline methods, we first turn the hyper-parameters to make models have the highest recommendation performance or highest parameter reduction rate. Then we tune those methods that can balance those two aspects. We provide the experimental details of our implementation for these baseline methods as below, following the settings of the original papers. For the grid search space of MDE, we search the baseline dimension $d$ from $[4, 8, 16, 32]$, the number of blocks $K$ from $[8, 16]$, and $\alpha$ from $[0.1, 0.2, 0.3]$. For MGQE, we search the baseline dimension $d$ from $[8, 16, 32]$, the number of subspace $D$ from $[4, 8, 16]$, and the number of centroids $K$ from $[64, 128, 256, 512]$. For DartsEmb, we choose three different candidate embedding spaces to meet the different memory budgets: $\{1, 2, 8\}, \{2, 4, 16\}$ and $\{4, 8, 32\}$.

### A.5 COMPARISON BETWEEN PEP-0 AND LINEAR REGRESSION

The Linear Regression (LR) model is an embedding-free model that only makes predictions based on the linear combination of raw features. Thence, it is worth comparing our method on the extremely-sparse level (PEP-0) with LR.

Table 4 shows that our PEP-0 significantly outperforms the LR in all cases. This result verity that our PEP-0 does not depend on the LR part in FM and DeepFM to remain a strong recommendation performance. Therefore, even at an extremely-sparse level, our PEP still has high application value in the real-world scenarios.

Table 4: Performance comparison between PEP-0 and Linear Regression.

| Methods | MovieLens-1M | | Criteo | | Avazu | |
|---|---|---|---|---|---|---|
| | AUC | # Param | AUC | # Param | AUC | # Param |
| LR | 0.7717 | 0 | 0.7881 | 0 | 0.7499 | 0 |
| PEP-0 (FM) | 0.8368 | 6,541 | 0.7941 | 1,067 | 0.7598 | 1,479 |
| PEP-0 (DeepFM) | 0.8491 | 8,604 | 0.7986 | 1,227 | 0.7622 | 2,215 |
| PEP-0 (AutoInt) | 0.8530 | 9,281 | 0.7922 | 3,116 | 0.7607 | 2,805 |
| PEP-0 (AutoInt+LR) | - | - | 0.7980 | 1,117 | 0.7620 | 2,225 |

It is worth noting that the AutoInt model does not contain the LR component, so the PEP-0 in AutoInt on Criteo and Avazu dataset lead to a large performance drop. We try to include LR in PEP-0 in AutoInt and test the performance[13]. As we can see, the accuracy on Criteo and Avazu outperforms the AutoInt without LR; It can be explained that LR helps our PEP-0 acquire a more stable performance.

## A.6    THE LOTTERY TICKET HYPOTHESIS

In the retraining stage in Section 4.2, we rely on the Lottery Ticket Hypothesis to reinitialize the pruned embeddings table (called winning ticket) into their original initial values. Here we conduct experiments to verify the effectiveness of this operation in our PEP. We compare our method with its variation that uses random re-initialization for retraining to examine the influence of initialization. We also compare the standard PEP with the original base recommendation model to verify the influence of embedding pruning. To evaluate the importance of retraining, we further test the performance of PEP with the pruning stage only. We choose FM as the base recommendation model and use the same settings as the above experiments.

We present the results in Figure 8 and 9. We can observe that the winning ticket with original initialization parameters can make the training procedure faster and obtain higher recommendation accuracy compared with random re-initialization. This demonstrates the effectiveness of our design of retraining. Moreover, the randomly reinitialize winning ticket still outperforms the unpruned model. By reducing the less-important features' embedding parameters, model performance could benefit from denoising those over-parametered embeddings. This can be explained that it is likely to get over-fitted for those over-parameterized embeddings when embedding sizes are uniform.

Moreover, it is clear that the performance of PEP without retraining gets a little bit downgrade, but it still outperforms the original models. And the margin between without retrain and the original model is larger than the margin between with and without retraining. These results demonstrate that the PEP chiefly benefits from the suitable embedding size selection. We conjecture the benefit of retraining: during the search stage, less-important elements in embedding matrices are pruned gradually until the training procedure reaches a convergence. However, in earlier training epochs when these elements have not been pruned, they may have negative effects on the gradient updates for those important elements. This may make the learning of those important elements suboptimal. Thus, a retraining step can eliminate such effects and improve performance.

## A.7    PRUNING WITH FINER GRANULARITY

In this section, we analyze the four different thresholds with different granularity mentioned in Section 4.3. The experiments are conducted on the MovieLens-1M dataset with base model FM. Figure 10 (a) and (b) demonstrates the varying of embedding parameters and test AUC evolving with training epoch. As we can see, the Feature-Dimension granularity can reduce much more embedding parameters than others. Meanwhile, it achieves the highest performance at the retrain stage compared with other granularities. With the minimum granularity, the Feature-Dimension wise pruning can effectively determine the importance of embedding values. Besides, the Dimension-wise prun-

---

[13]We omit the results of AutoInt with LR on the MovieLens-1M dataset because there is no performance drop for the AutoInt model compared with other models.

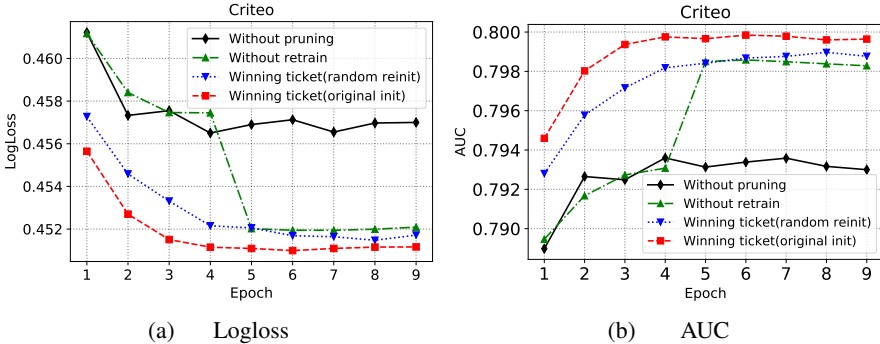

Figure 8: Logloss and AUC as training proceeds on Criteo dataset (choosing FM as the base model).

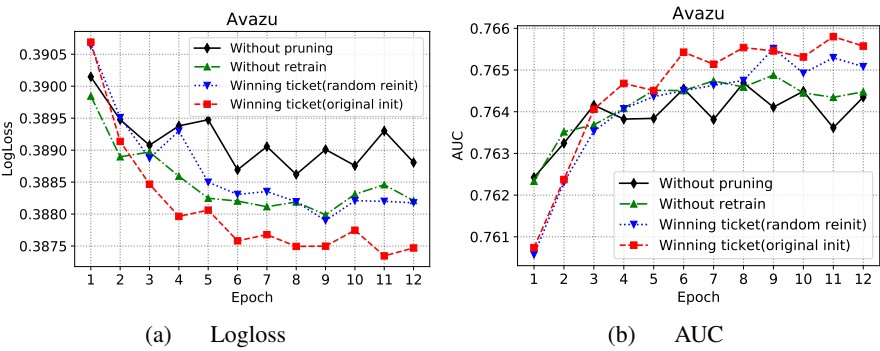

Figure 9: Logloss and AUC as training proceeds on Avazu dataset (choosing FM as the base model).

ing can achieve comparable AUC with fewer training epochs. Hence we adopt this granularity on PEP-2, PEP-3, and PEP-4 in large datasets to save time spent on training.

## A.8 ABOUT LEARNABLE $g(s)$

Pruning threshold(s) $g(s)$ can be learned from training data to reduce parameter usage in the embedding matrix. However, why can our PEP learn suitable $g(s)$ with training data? We deduce that the increase of $s$ in $g(s)$ can decrease the training loss. In other words, our PEP tries to update $s$ in the optimization process to achieve lower training loss.

In Figure 11, we plot the FM's training curves with/without PEP on MovieLens-1M and Criteo datasets to confirm our assumption. Our PEP can achieve much lower training loss when pruning. Besides, it verifies that our PEP could learn embedding sizes in a stable form.

The stability shown in Figure 11 can be explained that our PEP obtains a relatively stable embedding parameter number at later stage of pruning (e.g., when epoch is larger than 30 in MovieLens dataset) as shown in Figure 11. And embedding parameters are well-trained. Thus, the training loss curve looks relatively stable. Note that the figure shows a sequence of changing thresholds. The point when we get the embedding table for some sparsity level is not a converged point for this exact level, which instead requires retraining with a fixed threshold.

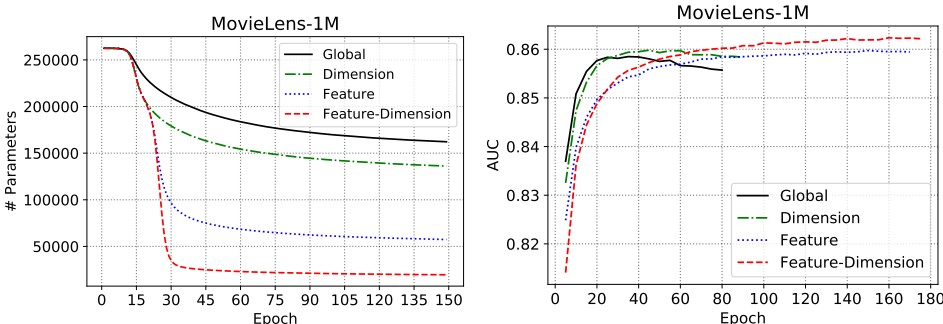

(a) Numbers of embedding parameters evolving with training epoch increase

(b) Test AUC evolving with training epoch increase at retrain stage

Figure 10: Influence of different granularity on MovieLens-1M dataset (Choose FM as base model)

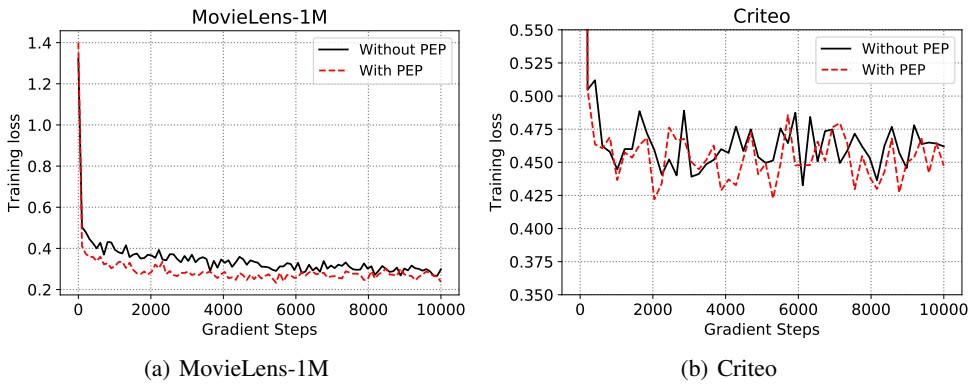

(a) MovieLens-1M

(b) Criteo

Figure 11: Training loss of FM with/without PEP

