# OpenReview forum: "Learnable Embedding sizes for Recommender Systems"
_ICLR.cc/2021/Conference — ICLR 2021 Poster_

### Official Review · AnonReviewer3 · 2020-10-27
**Great results**

**Rating:** 7
**Confidence:** 4

**Review:**

The paper investigates the embedding compression problem for recommendation via pruning. The paper is written very clearly with very strong experiments (multiple datasets/recsys models/recent baselines), showing ~99% parameter reduction while maintaining the same or even better performance.

I have a few questions:

- It's a bit surprising to me this method works. Like NIS or AutoEmb, they choose to use AutoML or validation data to select a suitable dim, as merely relying on a training set would lead to zero compression (using all parameters will have the lowest training loss). The key eq 8. actually doesn't encourage the model for sparsity, right? It's not clear to me why the model will increase `s`  during training, as using all parameters should decrease the training loss faster. Any explanation?

- Ablation study: it'd be good to see the effect of the sparsity constraint. That is to say, we may train a model without any modification, prune small values, and then re-train the model, basically following the Lottery Ticket paper. Via showing the performance of this baseline, we'd know the effect of the eq 8. It's not clear to me the better performance is from eq8 or the lottery ticket.

- What's the performance without re-training the model? Is it necessary to retrain the model for a good performance? What's the possible reason behind this?

- It'd be better to mention that one potential drawback of this method against the baselines is the sparse embedding table may not be very easily used in practice. If we store the full dense embedding matrix, we won't save the space. If we use sparse representation, it'd be slower for matrix operations. In contrast, baselines like NIS, are free of such an issue, as they have the same, smaller dimension for all feature values.

In summary, the results seem to be very promising, which is the most important reason for me to give a high rating. However, it's not clear to me why such a simple method works (doesn't rely on validation set, automatically prunes itself with training set only) very well, and a few ablation study experiments might be added. It's likely that I'd change my score based on the author's reply, e.g. I'd be happy to raise the score if the authors provide a clear explanation/intuition for the idea.

[updated after discussion]
Thank the authors for their efforts to add ablation study and make the manuscript more clear in presentation, it greatly resolves my questions and thus I raised the score. Overall the paper is in a good shape now. However I do want to point out a couple of things: 1) the performance of UE in base model FM/DeepFM/AutoInt seems to be a bit weired, as in previous CTR paper, deep models should outperform FM significantly. It's not clear to me is this due to different experimental settings or training schemes. 2) if the target is better performance (instead of compression), there is no clever way to choose s, other than mannual picking for each dataset/model.

---

> ### Public Comment · ~Aditya_Kusupati1 · 2020-11-11
> **Answer to the first question and waiting for the response for 2nd and 3rd.**
>
> Hi,
>
> That is a legitimate question to have. This paper is a practical application of STR [1] on the large embedding matrices of recommender systems. The dynamics for why the sparsity is induced are provided in detail in STR paper along with examples of the trajectory and the update equations. Check Figure 1, 3 Appendix A.1 and A.2 and Section 3.
>
> I also have the question on how does it work without re-training and want to see the ablation of using simple magnitude pruning.
>
> Thanks
>
> [1] Soft Threshold Weight Reparameterization for Learnable Sparsity, Kusupati et al, ICML'20.

---

> > ### Author Response · Authors · 2020-11-18
> > **Response to the public comment**
> >
> > Thanks for your attention and comments on our work. We have updated the results of PEP without retraining. The results demonstrate that the PEP benefits firstly from the suitable embedding size selection and secondly from the Lottery Ticket Hypothesis. Please check our response to the 3rd question of AnonReviewer3 for details. We also have updated the results about the ablation of using simple magnitude pruning in A.7 Pruning with Finer Granularity, in which we name the simple magnitude pruning as "Global". Obviously, with maximum granularity, global threshold pruning achieves the lowest pruning rate and the worst model performance among four kinds of pruning granularity settings.

---

> ### Author Response · Authors · 2020-11-18
> **Response to AnonReviewer3**
>
> Thanks for the constructive comments, and we hope the following answers can be useful for addressing your concerns.
>
> **Q1**: It's not clear to me why the model will increase $s$ during training.
>
> **Response**: As shown in Section 5.1, 5.3, and A.6 (revised version), our PEP can improve the recommendation performance via pruned flexible embedding sizes, compared with reserving all embedding parameters. Therefore, the assumption that a larger embedding size must lead to better fitting on the training set may not apply here. That is, besides the embedding matrices, the optimization of $s$ can also reduce the Logloss during the training; thus, a suitable "s" can be learned from training data.
>
> **Q2**: It's not clear to me the better performance is from Eq8 or the lottery ticket.
>
> **Response**: In Section A.6, The lottery ticket hypothesis, we compared the model performance of 1) retraining the model with the LTH and 2) retraining the model without the LTH. From the results, we conclude that both Eq(8) and LTH contribute to better performance.
>
> **Q3**: What's the performance without retraining the model.
>
> **Response**: We have updated Section A.6 to show the AUC and Logloss of our PEP without retraining. We can observe that the performance gets a bit worse, but it still significantly outperforms the base models. Besides, the margin between the base model and PEP-with-retrain is much larger than the margin between PEP-with-retrain and PEP-without-retrain. These results demonstrate that our PEP benefits chiefly from the suitable embedding size selection and secondly from the Lottery Ticket Hypothesis. The retraining step is inspired by [1], and this paper also demonstrated that this retraining step is critical to recovering the model's performance. The benefit of retraining in PEP can be explained as follows. During the search stage, less-important elements in embedding matrices are pruned gradually until the training procedure reaches a convergence. However, in earlier training epochs when these elements have not been pruned, they may have negative effects on the gradient updates for those important elements. This may make the learning of those important elements suboptimal. Thus, a retraining step can eliminate such effects and improve performance.
>
> **Q4**: Sparse embedding table may not be very easily used in practice.
>
> **Response**: Our work mainly focuses on reducing the parameter usage while remaining or even boosting the recommendation performance. The experiments show a significant reduction of parameter size (97%-99%), and we also observe that nearly 90%-99.9% of features are pruned entirely on all dimensions and can be represented as a zero vector. Thus, the significant reduction of embedding matrices makes the slightly-lower sparse matrix computation a minor issue. Even so, the storage cost is still affordable if we store embeddings in normal format rather than sparse format since a lot of feature embeddings are pruned, and only from hundreds to tens of thousands of features are reserved. Besides, it is worth mentioning that the acceleration of sparse matrix multiplication has been well explored by researchers and engineers, such as SparseX [2] and CUDA [3].
>
> ----
> Refs:
> [1] Song Han et al. Learning both weights and connections for efficient neural networks. In NeurIPS, 2015.
>
> [2] Elafrou A et al. SparseX: A Library for High-Performance Sparse Matrix-Vector Multiplication on Multicore Platforms. In ACM Transactions on Mathematical Software (TOMS), 2018.
>
> [3] Bell N et al. Efficient sparse matrix-vector multiplication on CUDA. Nvidia Technical Report NVR-2008-004, Nvidia Corporation, 2008.

---

> > ### Comment · AnonReviewer3 · 2020-11-18
> > **Response**
> >
> > Thanks for the work for improving the manuscript, it's pretty helpful for solving my concerns. I only have two more comments/questions:
> >
> > 1, in Q1, it seems there is no experiments in terms of training loss, so it still remains unclear whether introducing s and pruning could lower the training loss as you argued.
> >
> > 2, how do you get models with different sparsity: PEP-0, PEP-1, etc.? After the 1st training, I assume you only prune the model based on s, then you will only have one variant? or you choose other values? This is unclear to me.

---

> > > ### Author Response · Authors · 2020-11-24
> > > **Response to AnonReviewer3's New Comment**
> > >
> > > Thanks for your kind feedback, and we hope our answers are useful.
> > >
> > > **1.** Experimental evidence about that $s$ can increase during training.
> > >
> > > **Response**:
> > >
> > > During the training, the experimental evidence of the learnable $s$ is shown in Figure 10 (a). Since $s$ itself is hard to visualize, we present the parameter-number's descending trajectory which is directly caused by $s$'s ascending trajectory. This means, during the gradient-based learning, the loss function gradient with respect to $s$ can increase $s$.
> > >
> > > Furthermore, we present the curve of training loss in Figure 11 in the newly-added Section A.8. We can observe that PEP's training curve has a similar pattern to the base model. The training loss of PEP is close to or even lower than the base model. It is interesting that a model with smaller embeddings sizes can have lower training loss compared with a larger model. This might be explained that a model with large embedding sizes is more expressive but not necessarily easy to optimize. For recommender system models (FM, DeepFM, AutoInt, etc.), which is very different from CV and NLP, the optimal solutions can be very sparse — many feature interactions can be useless. If we don't introduce $0$ explicitly, the numerical optimization process might not find a stable sparse solution and instead pick a suboptimal dense solution.
> > >
> > >
> > > **2.** How to get models with different sparsity: PEP-0, PEP-1, etc.?
> > >
> > > **Response**：At the pruning stage, embeddings’ parameter number will decrease gradually. We first choose a set of target parameters $\mathcal{P} = \{P_0, P_1, ..., P_n\}$, where $P_0 < P_1 < ... < P_n$. If the parameter number of embeddings decreases to $P_n$, our PEP will save the current pruned embedding table, and the model will keep training until it reaches the smallest target parameter $P_0$. Finally, we can obtain several sparse embedding tables with different memory budgets, named PEP-0, PEP-1, ..., PEP-n. This reflects our PEP’s advantage--we obtain several pruning manners in one training instance.

---

> > > > ### Comment · AnonReviewer3 · 2020-11-24
> > > > **Question**
> > > >
> > > > Thanks for the reply, it makes things much clearer. However, I still have a question regarding the training dynamic.
> > > >
> > > > In Figure 8 (I assume Figure 8 is for testing results) and Figure 11 (for training loss), you showed that through the training, the PEP performance gradually goes better (and stable). However, you also said, "parameter number will decrease gradually" and thus lead to more compact models and worse performance (in Figure 4, PEP-0, PEP-1, etc.)
> > > >
> > > > The above facts are **contradicted** to me: the former suggests, through the training, the peformance goes better and better. while the latter suggests, through the training, the model prunes itself more and more, and thus the performance becomes worse.
> > > >
> > > > I think this is a key question for me to understand the paper, and hope you could clarify. Thanks!

---

> > > > > ### Author Response · Authors · 2020-11-25
> > > > > **Response to AnonReviewer3's New Question**
> > > > >
> > > > > **Question**: Contradiction of Figure 4, 8 and 11.
> > > > >
> > > > > **Response**:
> > > > >
> > > > > As we have mentioned, as the searching procedure continues, $s$ keeps increasing and embeddings are pruned gradually. We set a list of different sparsity demands and save corresponding $s$ during the searching procedure. Then we have different variants of PEP in Figure 2/3/4.
> > > > >
> > > > > To answer your question, we take PEP-i and PEP-j as an example. In Figure 10, assume we obtain $s_i$ of PEP-i at I-th epoch and $s_j$ of PEP-j at J-th epoch (I<J). In Figure 11, PEP-i’s pruning manner ($s_i$) only appears once at the moment of the I-th epoch. The performance at this moment cannot represent PEP-$i$’s final performance because it **has not converged** at the initial step of pruning. In Figure 2/3/4, the results show the converged  testing performance via full training with a fixed $s_i$.
> > > > >
> > > > > The stability shown in Figure 11 can be explained that our PEP obtains a relatively stable embedding parameter number when the epoch is larger than 30, as shown in Figure 10 (please note that Figure 11 and Figure 10 use different x-axis). And embedding parameters are well-trained. Thus, the training loss curve looks relatively stable.
> > > > >
> > > > >
> > > > > In summary, the contradiction is only due to our unclear presentation and we have updated the paper to claim there is no contradiction.

---

### Official Review · AnonReviewer1 · 2020-10-28
**A simple, yet effective technique PEP that proposes an LTH style iterative embedding pruning algorithm with learnable threshold for sparsity generation.**

**Rating:** 7
**Confidence:** 4

**Review:**

The paper proposes PEP (Plug-in Embedding Pruning) to reduce the size of embedding table while incurring insignificant drop in accuracy. The related work is well summarized into Embedding Parameter Sharing and Embedding Size Selection methods and the motivation for the current approach is well explained. The paper draws inspiration from Lottery Ticket Hypothesis. The problem formulation of Embedding pruning is done in a crisp way avoiding additional hyper parameter tuning that can be found in other methods. Similar to LTH, the paper shows that the initiation strategy can make the training process faster and stable. The results show an impressive 97-99% parameter pruning via PEP. As for the computation cost, PEP results show an additional 20-30% time cost compare with base models.

One of the main pros of this techniques is the learnable threshold(s) for pruning that can automatically jointly train with other parameters via gradient descent. As L0 based sparsity constraint on loss function is non-differentiable, the paper uses a neat re-parameterization technique to directly optimize the projection of sparsity matrix. The paper utilized the automatic differentiation framework to avoid complex back propagation and gradient calculation process. The main code of PEP is easy to read and understand. It’s only a few lines of code and well commented.

The results are well summarized and analyzed. Several baseline models (FM, DeepFM and AutoInt) were chosen that represented feature based models which was a fair strategy to follow. The one additional result that should have been added was running time comparison with MGQE, MDE, and DartsEmb pruning algorithms. The run time comparison of PEP was only shown with DartsEmb where the paper claims only 20-30% additional computation cost.

Post Pruning analysis on MovieLens Dataset was interesting and show that PEP assigns high- frequent features larger sizes and fewer parameters are enough for low-frequent feature embeddings. A missing portion which when added can make the paper wholesome is analysis of PEP across models for the same dataset. The results show that PEP performed best compared to other methods for FM model whereas for AutoInt model, AUC of PEP method was lower than MGDE for instance. Any hypothesis, intuition of why that would happen will add value to the paper.

---

> ### Author Response · Authors · 2020-11-18
> **Response to AnonReviewer1**
>
> Thanks a lot for your valuable comments and suggestions. Our responses are as below.
>
> **Q1**: Running time comparison with MGQE and MDE.
>
> **Response**: In MGQE and MDE, the embedding table's parameter number is determined by some hyper-parameters, as mentioned in Section 2 Related Works. Specifically, in MGQE, hyper-parameters determine the size of shared centroid embeddings; in MDE, hyper-parameters determine the proportional function between features' popularity and embedding sizes. These two methods choose proper hyper-parameters by grid-search, which means that each set of embedding size requires a full training procedure of the recommendation model. In CTR tasks, model training always costs the majority of the time. For our PEP and DartsEmb, the embedding size selection is coupled into the model-training procedure, which means the model only requires to be trained once. This significantly reduces the time cost.
> Here we also provide the experimental details of our implementation for these two methods, following the original papers' settings. For the grid search space of MDE, we search the baseline dimension $d$ from {4, 8, 16, 32}, the number of blocks $k$ from {8, 16}, and $\alpha$ from {0.1, 0.2, 0.3}. For MGQE, we search the baseline dimension $d$ from {8, 16, 32}, the number of subspace $D$ from {4, 8, 16}, and the number of centroids $K$ from {64, 128, 256, 512}. Thus, the searching cost is quite high. It is worth mentioning that these two methods' recommendation performance is still far worse than our PEP, although they spend far more time searching for suitable embedding sizes.
>
> **Q2**: Analysis of PEP across models for the same dataset.
>
> **Response**: Here, we provide an explanation for the performance drop of the AutoInt model with PEP-0 on large datasets (Criteo and Avazu): AutoInt model [3] does not contain a 1-order Linear Regression (LR) component, whereas FM [1] and DeepFM [2] both contain it. Specifically, LR's influence will be more significant since in an ultra-sparse situation (a lot of features embeddings are pruned), only relying on feature embeddings may make recommendations performance unstable. To verify this, we evaluate the performance of explicitly adding the LR component in AutoInt.
> We have added the experimental result in Table 4 at Section A.5, Comparison between PEP-0 and Linear Regression. As we can observe, for the accuracy of PEP-0, AutoInt with LR outperforms AutoInt without LR on both Criteo and Avazu datasets. Besides, AutoInt can achieve similar or better performance compared with FM and DeepFM. Hence we assume that adding the LR component to AutoInt helps our PEP-0 acquire a more stable performance. It is worth emphasizing again that our PEP-0 not merely depends on LR to obtain a good performance in an ultra sparse situation, which can be verified by the results (Table 4 in A.5) of comparison between PEP-0 and LR.
>
> ---
>
> Ref:
> [1] Steffen Rendle. Factorization machines. In ICDM 2010.
>
> [2] Huifeng Guo et al. Deepfm: a factorization machine based neural network for ctr prediction. In IJCAI 2017.
>
> [3] Weiping Song et al. Autoint: Automatic feature interaction learning via self-attentive neural networks. In CIKM 2019

---

### Official Review · AnonReviewer4 · 2020-10-30
**Interesting Application of Learnable Sparsity in Recommender Systems**

**Rating:** 7
**Confidence:** 3

**Review:**

In this paper, the authors apply the technique of 'learning' the sparsity structure [1] in the context of Neural Recommender Systems.

It's an interesting and very practical application of the technique in the context of RecSys. In designing large-scale RecSys models, where the number of distinct items and users can go in the range of millions, storing and retrieving can become a challenge. A parameter compression technique can indeed prove to be very effective in such situations.

Some strong points of the paper:

1) Provides a practical application of the technique described in the very recent ICML paper [1].
2) Paper is well-motivated and generally well-written.
3) Results indicate a huge improvement in terms of the number of parameters, without sacrificing the performance in the downstream tasks.

Some questions for the authors:
1) Can you please recheck equation 8? In the original paper, the equation is defined for each element of the matrix, where the mod. operator makes sense, you this case, what is the significance of the mod. the operator on the entire matrix?

Overall, given a very practical solution to the exploding users and items situation in a RecSys setting, I recommend an accept for the paper.






Ref:
[1] Soft Threshold Weight Reparameterization for Learnable Sparsity, Kusupati et al, ICML'20.

---

> ### Author Response · Authors · 2020-11-18
> **Response to AnonReviewer4**
>
> Thanks for taking the time to review our paper, and here we provide our response to your question about equation 8: In Eq(8), we want to convey that all elements in the embedding matrix will be conducted a pruning based on the same $g(s)$, so the Eq(1) in [1] can be adapted to Eq(8) in our paper.
>
> ---
> Ref:
> [1] Kusupati et al., Soft Threshold Weight Reparameterization for Learnable Sparsity. In ICML'20.

---

### Official Review · AnonReviewer2 · 2020-11-08
**The paper discussed how to make embedding vector dimension adaptive to model performance and memory cost in recommendation scenarios.**

**Rating:** 6
**Confidence:** 4

**Review:**

This paper proposed a novel approach to reduce size of the embedding table while not to drop in accuracy and computational optimization. Fixed-size embedding table has two problems, high memory usage cost and overfitting problem for those features that do not require too large representation. This paper recast the problem of embedding-size selection into learning column-wise sparsity, constraint K (eq(7)) and then convert S(V,s) problem (eq(8)). Paper used three benchmark datasets and some classical methods to verify effect.

Pros.
1.proposed a novel idea to learn embedding size  instead of fixed-size.
2.Experiments baseline with classical datasets and models show promising results on recommendation ranking problem.
3.The method has generalization ability across different recommendation models.

Cons
1.In METHODOLOGY part,  paper didn’t explain how to convert constraint by K in eq(7) to S(V,s) in eq(8)  and prove them.
2.Symbols in paper are confused such as V and v , and  N has different meanings in METHODOLOGY part.
3.The method first trains a sparse embedding matrix and then do re-training. During the sparse embedding learning phase, the memory size can not be reduced to a sparse format leading to significant memory cost, this is not feasible for very large scale sparse features for CTR task.

---

> ### Author Response · Authors · 2020-11-18
> **Response to AnonReviewer2**
>
> Thanks for your valuable comments, and we hope the following responses can address your concerns.
>
> **Q1**: Paper didn't explain how to convert constraint by $k$ in eq(7) to $S(V,s)$ in Eq(8) and prove them.
>
> **Response**: In Eq(7), $k$ represents the constraint on the total number of embedding parameters called $L_0$-norm. To solve its non-convexity, we convert the Eq(7) to an $L_1$-norm optimization process free from handling $k$. In Eq(8), we use $g(s)$ in $S(V, s)$ as the constraint. To be specific, if an element's value (absolute form) in the embedding table $V$ is smaller than the corresponding value of $g(s)$, the value will be set to zero. Hence, the $g(s)$ determines how many elements in $V$ will be pruned and plays the role of $k$ in Eq(7) with a relaxed form. In such a way, the Eq(7) turns to Eq(8).
>
> **Q2**: Symbols in paper are confusing.
>
> **Response**: Thanks for pointing it out. For your mentioned first point, we use the capital form of $V$ to denote the whole embedding table and the ordinary form of $v$ to denote a single embedding vector. This is simply because an embedding vector $v$ is contained in the whole embedding table $V$. For your mentioned second point, we have changed $N$ in Eq(6) to $|\mathcal{D}|$ that represents the number of training instances for a better representation.
>
> **Q3**: During the sparse embedding learning phase, the memory size can not be reduced to a sparse format leading to significant memory costs.
>
> **Response**: Most learning-based embedding size selection models, like NIS, DNIS, DartsEmb, and AutoEmb, are faced with this problem. When size selection begins, it is always required to assign a large enough embedding size to ensure the upper bound of representation ability. Besides, similar to the researches of neural network pruning [1, 2, 3], we mainly focused on reducing the parameter number, which can effectively reduce resource demands of neural network inference, including storage requirements, energy consumption, and latency [2].
>
> ---
> Refs：
> [1] Yann Le Cun et al. Optimal brain damage. In NeurIPS, 1990.
>
> [2] Song Han et al. Learning both weights and connections for efficient neural networks. In NeurIPS, 2015.
>
> [3] Zhuang Liu et al. Rethinking the value of network pruning. In ICLR, 2019.

---

### Author Response · Authors · 2020-11-18
**Response to all reviewers:**

We sincerely thank all the reviewers for their valuable feedback. Some statements, presentations, and experimental analysis in the original version are not clear and comprehensive enough. Thence we make the following revisions to the paper with the feedback from all reviewers:
1. In Section 1 Introduction, we have reorganized the 4-th paragraph to emphasize that both flexible embedding sizes and the Lottery Ticket Hypothesis benefit the recommendation performance.
2. In Section A.4.4, Implementation Details, we elaborate on the implementation of baselines.
3. In Section A.5, Comparison between PEP-0 and Linear Regression, we analyze why AutoInt's performances are worse than FM and DeepFM when applying PEP-0 in large datasets.
4. In Section A.6, The Lottery Ticket Hypothesis, we show the experimental results of PEP with only the pruning stage to verify the retraining stage's effectiveness.
5. In Section A.7, Pruning with Finer Granularity, we include the curve of pruning with maximum granularity, named global threshold pruning. The results further verify the effectiveness of finer granularity.
6. We have further checked our paper carefully to revise some minor issues like typos and unclear symbols to make the paper more readable.

----
New updates based on reviewers' new responses.


7. In Section A.8, About Learnable $g(s)$, we add the curve of training loss to further study the effect of $g(s)$.

---

### Decision · Program_Chairs · 2021-01-07
**Final Decision**

**Decision:**

Accept (Poster)

**Comment:**

This paper received overall positive scores. All the reviewers agree that the approach presented in the paper is simple yet effective and the results are very impressive with >95% parameter reduction while maintaining the accuracy. The authors promptly revised the paper based on initial reviews. Therefore, I recommend accept and hope the authors incorporate the additional comments from Reviewer3 after the discussion.